

# Micro-CT imaging in species description: exploring beyond sclerotized structures in lichen moths (Lepidoptera: Erebidae, Arctiinae, Lithosiini)

Simeão S. Moraes[1], Max S. Söderholm[2], Tamara M.C. Aguiar[1], André V.L. Freitas[1] and Pasi Sihvonen[2]

[1] Biologia Animal, Universidade Estadual de Campinas, Campinas, Brazil
[2] Finnish Museum of Natural History, University of Helsinki, Helsinki, Finland

## ABSTRACT

X-ray micro-computed tomography imaging (micro-CT) is valuable for systematic research since it permits the non-destructive scanning and imaging of internal structures of very rare species and/or type specimens. Additionally, micro-CT allows to view the morphology and the functional anatomy of structures in their natural anatomical position, without deformations that typically occur using classical dissection protocols. In this study we provide the description of two new species of lichen moths (Lepidoptera: Erebidae, Lithosiini) from the Atlantic Forest in eastern Brazil: *Nodozana heliae* Moraes **sp. nov.** from Rio de Janeiro state and *Epeiromulona pataxo* Moraes & Aguiar **sp. nov.** from Bahia state. The male and female genitalia as well as the wing morphology were examined by means of non-destructive micro-CT, subsequent 3D model reconstruction, 360 degree spinning animations, 2D images from different angles, and those were compared against classical genitalia dissections from the same specimens. We conclude that techniques complement each other, micro-CT being particularly useful to study wing venation, sclerotized internal structures and muscles, while classical dissection is useful to study membranous structures, particularly in the female genitalia, abdominal skin and specialised scales on the male 8th sternite.

## INTRODUCTION

Lepidoptera, commonly called butterflies and moths (or buttermoths), are covered with scales, tiny overlapping pieces of chitin that are derived from macrotrichia (*Kristensen & Simonsen, 2003*). The scales form a wide variety of diagnostic colors and patterns, particularly on wings but also on other body parts. For that reason, they are extensively used in Lepidoptera taxonomy and systematics as a source of characters for indentification (*e.g.*, color) and relationships among lineages (*e.g.*, ultrastructure) (*Kristensen & Simonsen, 2003*). The scales also hide many structural details, often relevant in systematics. These include wing venation, which is particularly important in suprageneric classification. Scales also hide modified sclerites, conventional setae, spike-like processes and secondary sexual

Corresponding author
Simeão S. Moraes,
simeao_moraes@yahoo.com.br

characters, including the eversible scent-producing coremata (*Scoble, 1995*; *Kristensen, 1999*; *Kristensen, 2003*). Other internal structures, which are important in systematics, include the reproductive organs and sclerites of the thorax, to mention a few. These hidden structures have traditionally been studied using various dissection techniques, usually resulting at least in the partial destruction of these (*e.g.*, *Hardwick, 1950*; *Sihvonen, 2001*; *Moraes & Duarte, 2009*; *Murillo-Ramos, Hernández-Mejía & Llorente-Bousquets, 2016*). Also, learning these dissection techniques requires extensive training so that only some experts can master this "craftsmanship" in such detail that museums allow the dissection of their valuable type or rare specimens.

As regards the genitalia, the dissection techniques such as potassium hydroxide (KOH) treatment (*Robinson, 1976*), if carefully applied, reveal the sclerotized structures very well, and those can be studied and imaged to the finest detail. The genitalia also contain membranous structures, which are more challenging to explore because they are transparent, delicate, and easily break or detach from other structures during the dissection process (this is the case, particularly in the female genitalia). A notable problem with the KOH treatment is that it readily dissolves the male-deposited spermatophores and female eggs inside the ovaries, or makes the muscles invisible, which means that important taxonomic information is lost. Additionaly, the KOH treatment is time sensitive, meaning that too long submerging of structures in the solution can dissolve the structures altogether. Staining enhances the study of the membranous structures. Still the problem remains that classical dissection methods damage some structures, and some of them are even removed routinely to make other structures visible. Finally, dissected structures are routinely stored in microvials with glycerin, or embedded in microscopic slides and mounted in an artificial position, such as the male valvae spread out. In some moth groups, the male genitalia are unrolled (*e.g.*, *Pitkin, 1986*), and the vesica is left uneverted inside the phallus. Some of these practises are useful for taxonomic research and for storing extensive materials in museums. Still, it does not allow the subsequent study of the structures from various angles, which are helpful in taxonomy and functional anatomy research.

The X-ray micro-computed tomography imaging (micro-CT) is a fast and non-destructive data acquisition technique that can complement traditional, partly desctructive dissection methods in morphological studies. Since its first application in entomology, 20 years ago (*Hörnschemeyer, Beutel & Pasop, 2002*), micro-CT is still relatively little used in insect morphology, mainly because the equipment is expensive, relatively few museums have it, and skilled imaging experts are few. The inspiring examples of micro-CT in Lepidoptera taxonomy include, among others, virtual dissections of reproductive organs (*Simonsen & Kitching, 2014*), wing venation study on over 200-hundred-year-old type specimen (*Robinson et al., 2018*) and functional morphology of internal structures (*Nath & Kunte, 2020*). Other benefits of the approach include the study of morphologies in their natural position without deformations, versatile post-processing of data (*Garcia et al., 2017a*), and the scrutiny of valuable material such as type specimens or scarce species without damaging the samples (*Stoev et al., 2013*; *Garcia et al., 2017b*). Moreover, this technique can also help to open novel avenues of research in 3D morphometrics analyses, for instance for wing venation, genitalia and tymbal organs.

The tribe Lithosiini, or Lichen moths, is a species-rich insect lineage whose subtribes and genus taxonomy are notorious, needing a modern integrative approach. The tribe includes approximately 3,150 species, classified into 457 genera (*Scott et al., 2014*). Of these, 345 genera are classified as *incertae sedis* (*Bendib & Minet, 1999*), and new species are described constantly (*e.g.,* *Durante & Apinda-Legnouo, 2022*; *Volynkin et al., 2022a*; *Volynkin, Cerny & De Vos, 2022b*). The tribe is exceptionally species-rich in the Neotropical region (*Chialvo et al., 2018*), and in Brazil, there are 212 species classified in 52 genera (*Moraes & Casagrande, 2019*). However, given the scarcity of specialists working with Neotropical fauna, these numbers are underestimated and potential existence of new records and species are expected.

While the morphology of Lithosiini moths has been studied extensively using classical methods (recent examples include *Durante & Apinda-Legnouo, 2022*; *Volynkin et al., 2022a*; *Volynkin, Cerny & De Vos, 2022b*), these moths have not been the target of micro-CT imaging. We chose two undescribed lichen moth species as our study species. These originate from the Brazilian Atlantic Forest, which is one of the Earth's Biodiversity hotspots with high levels of diversity and endemism (*Matos-Maraví et al., 2021*). The biodiversity and biomass in this area have been reported to be eroding at an alarming rate (*De Lima, Oliveira & Pitta, 2020*). We aim to explore how micro-CT imaging can enhance the study of morphological structures compared to classical dissections and how it can be applied in systematic research. Our study focuses on structures that are not directly visible, such as scale-covered wing venation and abdominal structures, and retracted and concealed genitalia, *in situ*, and finally, to conclude which of the two approaches (classical dissection or micro-CT imaging) is best suited for unveiling and imaging the fine morphology of different structures.

## MATERIAL AND METHODS

### Sampling and identification

Moth collecting was carried out in the Brazilian states of Rio de Janeiro and Bahia between 2016 −2021 during the new-moon phase to enhance light attractiveness using a 500 W mixed light bulb and a white 2 m × 2 m sheet to attract the moths. Specimens were individually kept alive in small glass containers and then were killed with ethyl acetate just before the wing spreading.

For identification, specimens were compared against relevant literature and online sources (*Seitz, 1914*; *Seitz, 1943*, taxonomy section on Barcode of Life Data Systems; https://v4.boldsystems.org/), to material in relevant collections (ZUEC, MZSP, ZMH), including both type and non-type specimens, and DNA barcodes (658 bp region near the 5′ terminus of the COI mitochondrial gene) were compared against the genetic material available on BOLD (*Ratnasingham & Hebert, 2007*; *Ratnasingham & Hebert, 2013*) and GenBank (*Benson et al., 2013*). Genetic divergences between sequences were calculated using the number of base differences between sequences. Voucher specimens are deposited at the institutions mentioned above (details are given under the *Species description* section).

## DNA extraction and PCR protocol

Three legs were removed from each specimen shortly after collection and before the wing spreading. Sampled legs were preserved dry and stored in 1.5 ml tubes at −20 °C. Total genomic DNA was extracted with QIAcube DNA extraction robot (Qiagen, Hilden, Germany) using DNeasy Blood & Tissue Kit standard protocol with final elution in 100 µl elution buffer. The 5′ end (barcode region) of the mitochondrial gene cytochrome oxidase subunit I (COI, 650 bp) was amplified for the SSM3 sample with LepF1 (*Wilson, 2012*) and HCO (*Folmer et al., 1994*) primers. The barcode for the SSM4 sample was amplified in two parts: the first half (∼330 bp) with LepF1 and mLepR1 (*Wilson, 2012*) primers and the end half (∼450 bp) with Beet (*Simon et al., 1994*) and HCO (*Folmer et al., 1994*) primers.

Polymerase chain reactions (PCRs) were performed with 13 µl total volume containing 3 µl of extracted DNA, 2 µl of $H_2O$ milli-Q, 6.5 µl of 2x MyTaq HS red mix (Bioline Co., London, UK), and 0.75 µl of each primer (10 mM). PCR products were amplified in a Bio-Rad PCR thermal cycler as follows: 96 °C for 7 min, followed by 40 cycles of 96 °C for 30 s, 50 °C for 30 s and 72 °C for 90 s, and a final extension period of 72 °C for 10 min. Amplicons were purified by mixing 5 µl of PCR product with 2 µl of 1/10 $H_2O$ milli-Q diluted ExoSAP-IT (Thermo Fisher Scientific, Waltham, MA, USA). Purification was run in a Bio-Rad PCR thermal cycler: 37 °C for 15 min and 80 °C for 15 min. Purified products were sent for Sanger sequencing to FIMM (Institute for Molecular Medicine Finland).

## Morphological examination

We first imaged the adult specimens using a non-destructive micro-CT approach and tried to identify external and internal morphological structures *in situ*. Following this, male and female abdomen and genitalia of both new species were dissected. The data from classical dissections an micro-CT allowed us to refine homology interpretations.

**Micro-CT imaging**. Each adult specimen was pinned using a minute pin, attached to a foam cube, traversed by standard insect pin. To avoid noise and artifacts resulting from the standard insect pin holding the foam cube (*Simonsen & Kitching, 2014*), the pin was pushed down to exclude it from the scanning area. The samples were imaged using Nikon XT H 225 micro-computed tomography. Scans were performed using a multi-metal target with a molybdenum setting, with 73–74 kV beam energy, 94–95 uA beam current, 500 ms exposure time, and 4,476 projections with four frames per projection, which resulted in an angular step of 0.0804 degrees and a total scan time of 2 h 29 m 18 s. Shading correction of 5 m 23 s and X-ray tube warm-up of 15 m was performed before the scans. Detector binning was set to 1×1 and gain to 24 dB. Imaging was conducted using limited dynamic range after performing comparisons that showed no visible differences between the longer full dynamic range scans and the faster limited dynamic range scans. Adequate pixel size ranged from 3.27 to 7.27 µm between the scans. Projections were processed using Nikon CT pro-3D, and the 3D models were exported to VGSTUDIO 3.5.2 (Volume Graphics GmbH, Heidelberg, Germany) in 16-bit. The 3D models were visualized using two renderer modes, volume renderer (Phong) and X-ray, and they were pseudo-colored to visualize density. Virtual section stacks in the three principal planes (coronal, sagittal, and axial) were exported in JPG format.

The external morphology and color pattern were analyzed following usual protocols (*Winter, 2000*; *Moraes & Duarte, 2009*).

**Dissection**. The abdomens and genitalia of females and males were dissected following standard methods (*Hardwick, 1950*; *Robinson, 1976*). The male phallus is shown with uneverted vesica to allow comparison with older literature and with everted vesica. The vesica was everted *via* the caecum cut open by placing the phallus inside a hypodermic syringe (*Sihvonen, 2001*). Some structures were photographed during dissection to allow an optimal angle for observing and illustrating specific structures. Numerous dissected structures shown in the plates were photographed in two to six images at different depths of focus, using a Leica DM1000 microscope and Leica DFC295 camera, and combined into single images using image-stacking software in Adobe Photoshop CC v.20.0. For interpretation and descriptions of the genitalia structures we followed the procedures outlined in *Moraes & Duarte (2015)*, and terminology of the male genitalia follows (*Pierce, 1909*; *Sibatani et al., 1954*; *Okagaki et al., 1955*; *Klots, 1956*; *Ogata et al., 1957*; *Birket-Smith, 1974*; *Moraes & Duarte, 2009*). Female genitalia follows (*Pierce, 1914*; *Klots, 1956*; *Mutuura, 1972*; *Galicia, Sánchez & Cordeiro, 2008*). The muscle nomenclature follows *Kuznetzov et al. (2004)*.

A total of 19 specimens belonging to the two new species were examined. Details are given under each species below.

The electronic version of this article in Portable Document Format (PDF) complies with the International Commission on Zoological Nomenclature (ICZN) rules. Hence, the new names in the electronic version are effectively published under that Code from the electronic edition alone. This published work and its nomenclatural acts have been registered in ZooBank, the online registration system for the ICZN (urn:lsid:zoobank.org:pub:68906FAC-208D-48D7-B69C-4ABDE6CFA0D6, urn:lsid:zoobank.org:act:F97C4D7C-65A3-4EEC-8D34-F84D5C7346EE, urn:lsid:zoobank.org:act:D4637E11-169E-45D0-8A25-6A61F68939A5). The ZooBank LSIDs (Life Science Identifiers) can be resolved and the associated information viewed through any standard web browser by appending the LSID to the prefix http://zoobank.org/. The online version of this work is archived and available from the following digital repositories: PeerJ, PubMed Central, and CLOCKSS. All the holotypes are deposited at Coleão Zoológica da Universidade Estadual de Campinas (ZUEC). The paratypes are deposited at Museu de Zoologia da Universidade de São Paulo (MZSP), the Finnish Museum of Natural History (ZMH) and ZUEC.

# RESULTS

## Micro-CT imaging

The micro-CT scanning and post-processing of 3D models allowed us to visualize clearly and in a non-destructive manner the wing venation and wing folds in both sexes (2D image in Fig. 1, 3D spinning video on Supplementary Material 1), several sclerotized structures in the male abdomen, and several scleroritized structures and muscles in the male genitalia (3D spinning video on Supplementary Material 2). The visible male structures include,

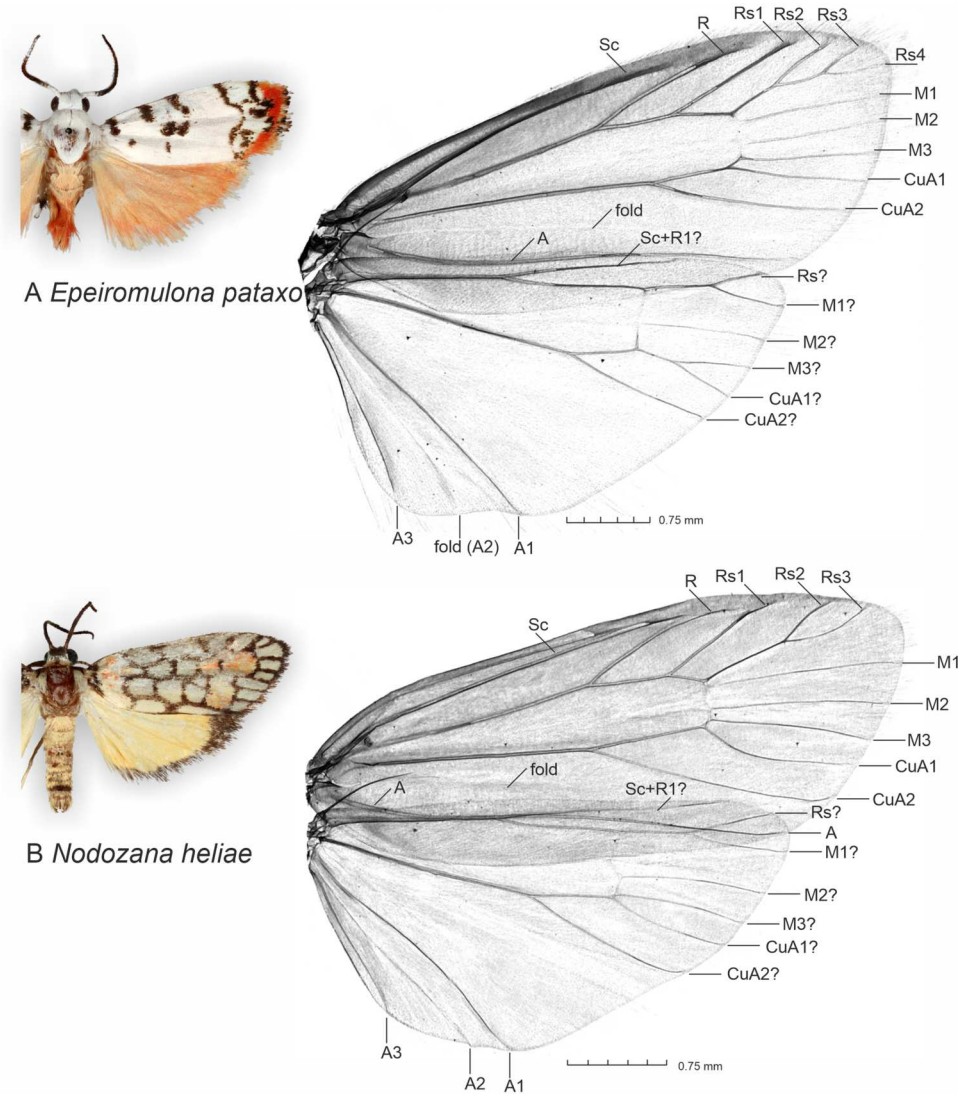

**Figure 1 General appearance of new described species and micro-CT images of wing venation.** (A) *Epeiromulona pataxo* Moraes & Aguiar **sp. nov**. (B) *Nodozana heliae* Moraes **sp. nov.** Venation terminology after *Wootton (1979)*.

for instance, the posterior margin of the 8th abdominal tergite, tegumen, uncus, valva, transtilla, phallus and cornuti (3D spinning video on Supplementary Material 2).

Genital muscles were visible in the 3D models (Supplementary Material 2). Those include

- the m1 muscle. The depressor of the uncus is a broad longitudinal muscle extending ventromedially from the anterior margin of the tegumen to the base of the uncus;
- the m5 (7) muscle. The flexor of the clasper is an intravalval muscle extending from the anterior part of the valva, in the region of the sacculus, to the central part of valva.

These muscles bend the valvae transversally, clasping the female abdomen during the copulation;

- the m6 (5) muscle. The protractors of the phallus originate on the dorsolateral region of the vinculum and insert on the base of the phallus;
- the m7 (6) muscle. The retractors of the phallus extend from the saccus and insert mid-ventrally on the phallus.

For the membranous structures embedded in the abdomen, the micro-CT scanning and post-processing of 3D did not produce clear images, and homology interpretation was difficult mainly because the density of membranous structures and body fat were similar. As regards the female genitalia, which are primarily membranous, most of the structures could be identified only *via* the sequential study of the sagittal slices (Supplementary Material 3 and 4). In the sagittal section, we were able to first detect the sclerotized signa, antrum, and ductus bursae, which we used as reliable morphological anchor points, and subsequently, from these images, it was possible to infer the outer surface of the membranous corpus bursae and ductus bursae.

Micro-CT imaging cannot be used to study the shape of eversible membranous structures. Among these is the male vesica, which we explored using the method of *Sihvonen (2001)*, demonstrating their complexity in both species.

## Species description

Comparison of our material, using both morphological and DNA barcode data, against described species of Lithosiini did not result in a positive match, thus suggesting our specimens belong to undescribed species. We provide the formal descriptions below.

*Nodozana heliae* Moraes **sp.nov.** (Figs. 1b, 2–3)

**Diagnosis** (♂ and ♀). Prothoracic collar orange. Dorsal surface of the forewing with several subrectangular white maculae, orange maculae on the wing base seahorse-shaped, orange maculae on the submarginal region hammer-shaped with an elliptical black spot inside. Compared with other *Nodozana* species, the wing pattern with white squares is idiosyncratic. Only *Nodozana toulgoeti* Gibeaux, 1983 has a similar wing pattern but without the red basal macula on the forewing.

**Description** (♂ and ♀). **Head**. Brown. Frons brown, vertex yellow. Labial palp brown. **Thorax**. Predominantly brown. Prothoracic collar orange; prothoracic coxa brown. Tegulae beige. **Wings**. Venation as in Fig. 1B: Wingspan 15.5 −17.5 mm ($n = 4$, three females, one male). Forewing background dark brown; basal orange seahorse-shaped macula; white subrectangular maculae distributed on post-basal, discal, post-discal and marginal regions; a post-discal region with orange hammer-shaped maculae between $M_1$ and CuP veins; elliptical black macula between $M_1$ and $M_3$; ventral surface of forewing dark brown, with a narrow orange stripe near wing apex; maculae obscured by dark brown scales. Hindwing yellow, apex with a dark brown macula, outer margin outlined by dark brown scales; ventral surface similar. **Abdomen**. Dorsally light brown, posterior margin of segments $A_2$-$A_8$ outlined by dark brown scales; ventrally similar. 7th sternite margins weakly sclerotized, posterior margin concave. 8th sternite with anterior and lateral margins sclerotized, setose coremata medially. **Male Genitalia**. Tegumen trapezoidal in dorsal view, anterior margin
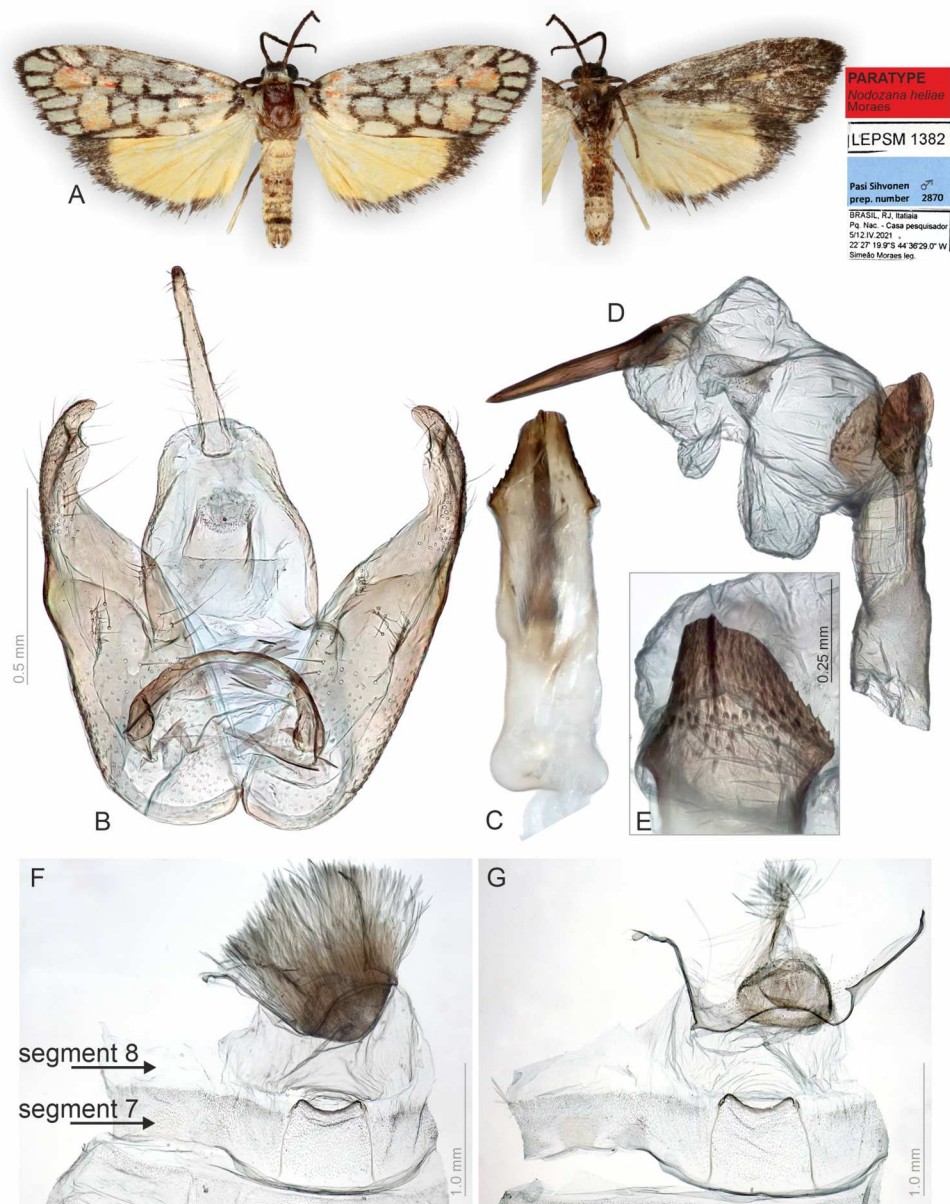

**Figure 2** **Habitus and male genitalia of *Nodozana heliae* Moraes sp. nov. (paratype).** (A) Habitus, dorsal and ventral view, wingspan 16 mm. (B) Genital capsule, ventral view. (C) Aedeagus, ventral view. (D) Aedeagus with everted vesica, lateral view. (E) Detail of micro spicules at apex of aedeagus. (F) 7th sternite and androconial scent organ associated to the 8th sternite. (G) 8th sternite with androconial scales removed. All pictures from dissection PMS2870.

concave. Uncus hooked; apex acute. Valva entire, sub-rectangular; sacculus developed, fold on the inner surface, oriented towards the distal-medial axis. Transtilla sclerotized, inverted U shaped. Juxta membranous. Subscaphium sclerotized. Phallus rectilinear, sclerotized, with micro spicules on the anterior portion near vesica; caecum rounded, foramen lateral; vesica large, with three large diverticula, single spiniform cornutus. **Female Genitalia**.

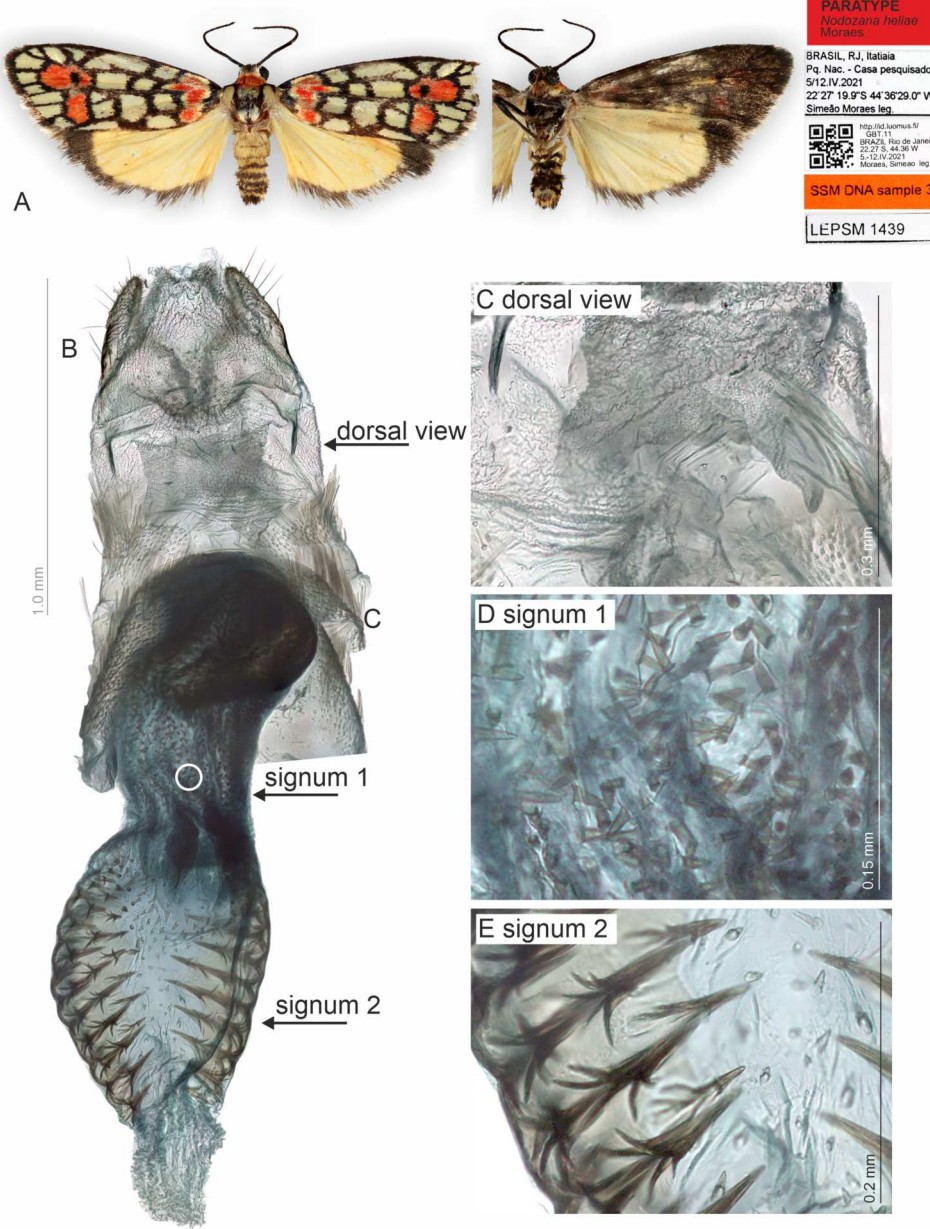

**Figure 3** **Habitus and female genitalia of *Nodozana heliae* Moraes sp. nov. (paratype).** (A) Habitus, dorsal and ventral view, wingspan 18 mm. (B) Female genitalia, ventral view, point of origin of ductus seminalis indicated with circle. (C) Detail of pheromone gland, dorsal view. (D) Detail of signum on posterior portion of corpus bursae. (E) Detail of signum on anterior portion of corpus bursae. All pictures from dissection SSM516.

Seventh sternite smooth. Ostium membranous. Antrum sclerotized with microspicules. Ductus bursae short, membranous. Corpus bursae extending beyond the seventh sternite, signa as two patches of spines on the lateral portion of the bursa. Lamella antevaginalis and postvaginalis absent. Papillae anales narrow, setose.

**Etymology**. The specific epithet is granted in honor of Maria Hélia de Souza Moraes, mother of the first author. It is also a reference between the golden scales on the forewing and Helios, the sun's personification in Greek mythology.

**Distribution**. The only record for this species is from the National Park of Itatiaia, a montane-dense ombrophilous forest of medium and high altitudes in Rio de Janeiro State, Brazil. The specimens were sampled in low altitude (below 1000 m).

**DNA barcode data**. ($n = 1$) from Brazil: Rio de Janeiro. Based on nucleotide blast function on GenBank, the nearest lichen moth species is *Nodozana toulgoeti* from French Guiana, with a 7% difference.

**Type series**. HOLOTYPE ♀: BRAZIL: **Rio de Janeiro**: Itatiaia, Parque Nacional do Itatiaia, Alojamento 12, 01-04/viii/2016, Simeão Moraes, Tamara Aguiar, André Taciolli leg., label: LEPSM 551 (ZUEC). PARATYPES: BRAZIL: 1 ♀ **Rio de Janeiro**: Itatiaia, Parque Nacional do Itatiaia, Alojamento 12, 01-04/viii/2016, Simeão Moraes, Tamara Aguiar, André Taciolli leg., labels: Simeão Moraes Genitalia 516, LEPSM 1093 (ZUEC); 1 ♂ **Rio de Janeiro**: Itatiaia, Parque Nacional do Itatiaia, Casa do Pesquisador 12, 05-12/iv/2021, 22°27′19.9″S 44°36′29″W, Simeão Moraes leg., labels: Pasi Sihvonen Prep. Number 2870, LEPSM 1382 (ZUEC); 1 ♀ **Rio de Janeiro**: Itatiaia, Parque Nacional do Itatiaia, Casa pesquisador, 05-12/iv/2021, 22°27′19.9″S 44.36′ 20.0″W, Simeão Moraes leg., labels: LEPSM 1439, SSM DNA sample 3, specimen ID http://id.luomus.fi/GBT.11 (ZMH).

*Epeiromulona pataxo* Moraes & Aguiar **sp.nov.** (Figs. 1a, 4–5)

**Diagnosis** (♂ and ♀). Forewing dorsally white, with several small black maculae on the proximal portion, and outer margin with reddish scales. Hindwing uniform salmon colored. The forewing pattern with white/beige background and black maculae/dots are shared with other *Epeiromulona* species, but the reddish outer margin is idiosyncratic for *Epeiromulona pataxo*.

**Description** (♂ and ♀). **Head**. White. Frons brown, vertex white. Labial palp white. **Thorax**. Predominantly white. Prothoracic collar white; prothoracic coxa white. Tegulae white. **Wings**: Venation as in Fig. 1A. Wingspan 12,75 −13 mm ($n = 15$, 14 males, one female). Forewing background white with two proximal maculae: elliptical on costal margin, rounded on the trunk of A vein; four maculae on the medial region: two elliptical on costal margin, subrectangular at the discal cell, subretangular at A vein; a submarginal region with two sinuous stripes, proximal longer than distal; outer margin with reddish scales; ventral surface reddish with black maculae fused along the basal length of costal margin, black stripes fused on the subapical region. Hindwing dorsally salmon-colored; ventrally salmon with apical black macula on the region of Rs and M1. **Abdomen**. Dorsally salmon-colored on $A_2$-$A_4$, reddish on $A_3$-$A_8$; ventrally salmon. Male segments 7–8 are not differentiated. **Male Genitalia**. Tegumen subrectangular in dorsal view, anterior margin concave. Uncus hooked, with acute apex. Vinculum narrow. Valva relatively immobile, setose, sub-triangular, apex rounded; sacculus developed, consisting of the fold on the

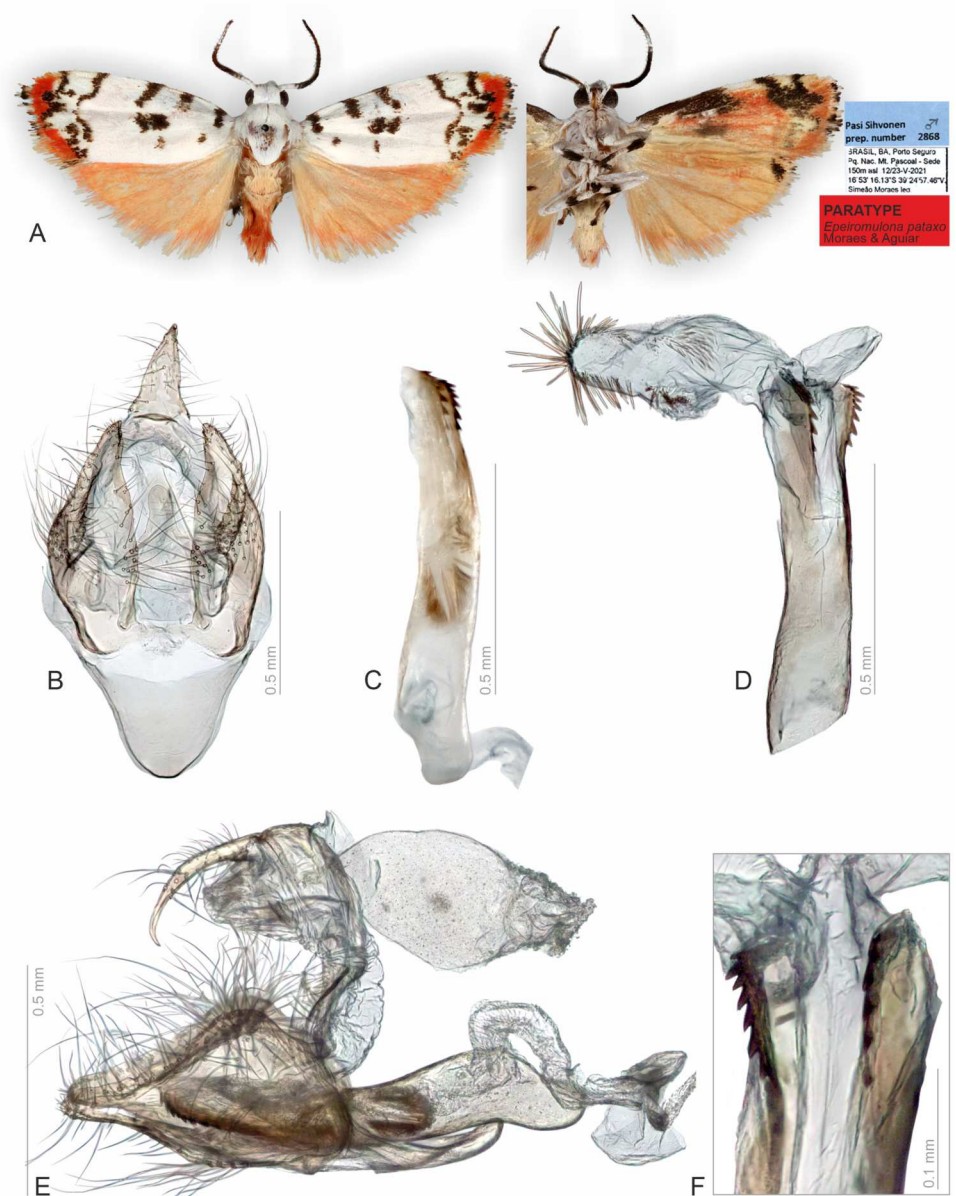

**Figure 4** **Habitus and male genitalia of *Epeiromulona pataxo* Moraes & Aguiar sp. nov. (paratype).**
(A) Habitus, dorsal and ventral view, wingspan 13 mm. (B) Genital capsule, ventral view. (C) Aedeagus, lateral view. (D) Aedeagus with everted vesica, lateral view. (E) Genital capsule with aedeagus, lateral view. (F) Detail of micro spicules at apex of aedeagus. All pictures from dissection PMS2868.

inner surface, oriented towards the distal-medial axis. Saccus large, subtriangular. Juxta sclerotized, subtriangular. Subscaphium smooth. Phallus rectilinear, sclerotized, two rows of spines on apex; ejaculatory bulb rounded, foramen lateral; vesica bilobated, cornuti on the larger lobe: micro spines medially and needle-shaped spines on distal part. **Female Genitalia**. Seventh sternite smooth. Ostium membranous. Antrum slightly sclerotized, smooth. Ductus bursae short, membranous. Corpus bursae massive, extending beyond the

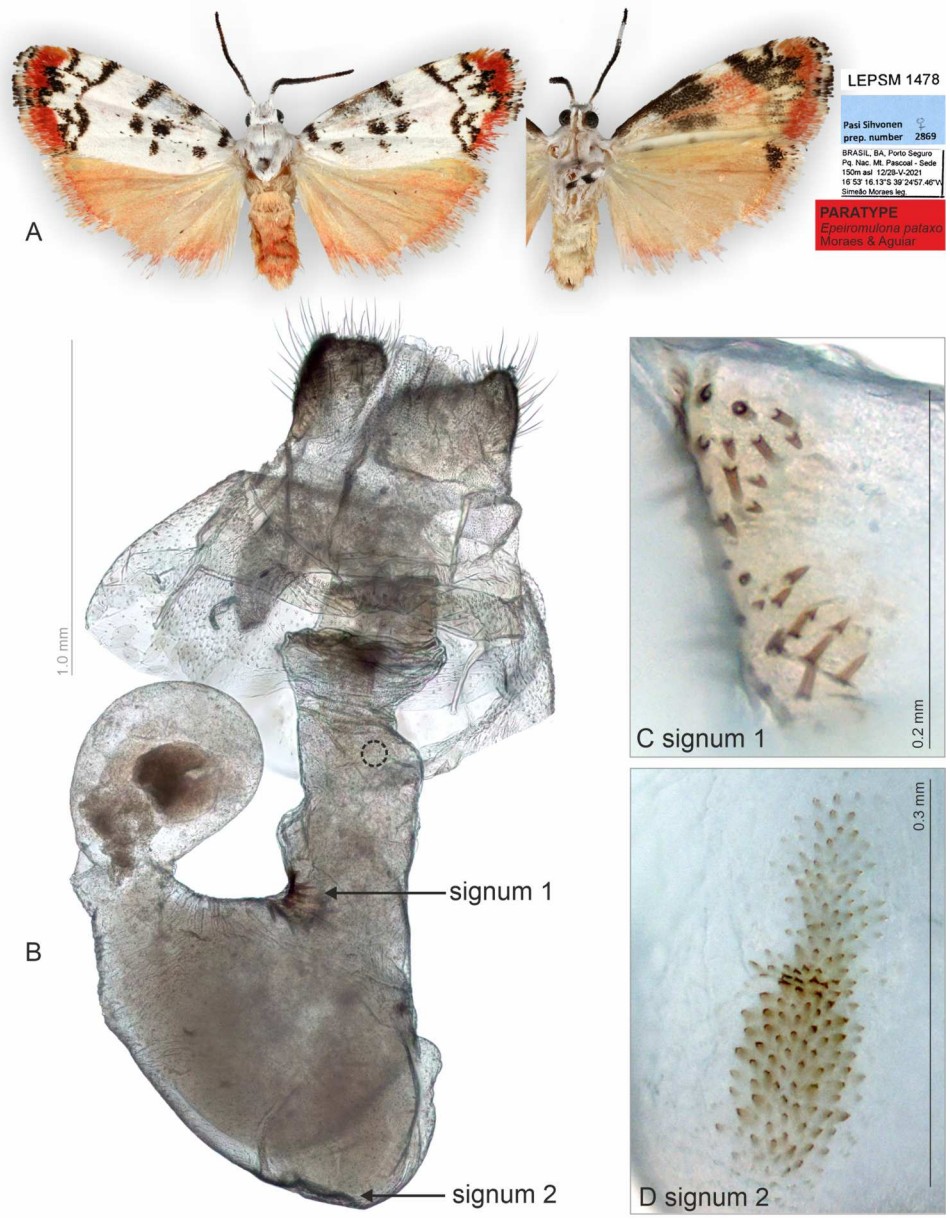

LEPSM 1478

Pasi Sihvonen
prep. number    2869

BRASIL, BA, Porto Seguro
Pq. Nac. Mt. Pascoal - Sede
150m asl 12/28-V-2021
16°53'16.13"S 39°24'57.46"W
Simeão Moraes leg.

**PARATYPE**
*Epeiromulona pataxo*
Moraes & Aguiar

**Figure 5** **Habitus and female genitalia of *Epeiromulona pataxo* Moraes & Aguiar sp. nov. (paratype).**
(A) Habitus, dorsal and ventral view, wingspan 14 mm. (B) Female genitalia, ventral view (point of origin
of ductus seminalis indicated with circle). (C) Detail of signum on posterior portion of corpus bursae. (D)
Detail of signum on anterior portion of corpus bursae. All pictures from dissection PMS2869.

seventh sternite, with two patches of signa consisting of spines on the posterior portion,
and micro spicules on the anterior portion. Lamella antevaginalis and postvaginalis not
sclerotized. Papillae anales large, setose.

**Etymology**. The specific epithet honors the Pataxós, indigenous people inhabiting the state of Bahia and the north of Minas Gerais State, in Brazil, which is the location where the specimens were collected. A masculine name in apposition.

**Distribution**. The only record for this species is the National Park of Monte Pascoal in Porto Seguro, State of Bahia, Brazil. The region represents one of the last remnants of the Atlantic Forest, where the predominant vegetation is a tall tropical rainforest whose physiognomic and structural aspect of its the dense and exuberant vegetation earned it the name of Hileia Baiana.

**DNA barcode data**. ($n = 1$) from Brazil: Bahia. Based on nucleotide blast function on GenBank, the nearest species is *Epeiromulona* sp. from Costa Rica, with a 7% difference.

**Type series**. HOLOTYPE ♂: BRAZIL: **Bahia**: Porto Seguro, Parque Nacional Monte Pascoal, Sede, 150 m, 12–23/v/2021, 16°53′16.13″S 39°24′57.46″W, Simeão Moraes leg., (ZUEC). PARATYPES BRAZIL: 1 ♀ **Bahia**: Porto Seguro, Parque Nacional Monte Pascoal, Sede, 150m, 12-23/v/2021, 16°53′16.13″S 39°24′57.46″W, Simeão Moraes leg., (ZUEC); 7 ♂ ♂ **Bahia**: Porto Seguro, Parque Nacional Monte Pascoal, Sede, 150 m, 12-23/v/2021, 16°53′16.13″S 39°24′57.46″W, Simeão Moraes leg., (ZUEC); 2 ♂ ♂ **Bahia**: Porto Seguro, Parque Nacional Monte Pascoal, Sede, 150 m, 12–23/v/2021, 16°53′16.13″S 39°24′57.46″W, Simeão Moraes leg., (MZUSP); 2 ♂ ♂ **Bahia**: Porto Seguro, Parque Nacional Monte Pascoal, Sede, 150 m, 12-23/v/2021, 16°53′16.13″S 39°24′57.46″W, Simeão Moraes leg.. 1 ♂ **Bahia**: Porto Seguro, Parque Nacional Monte Pascoal, Sede, 150 m, 12–23/v/2021, 16°53′16.13″S 39°24′57.46″W, Simeão Moraes leg., Genitalia SSM A, specimen ID http://id.luomus.fi/GBT.12 (ZMH); 1 ♂ **Bahia**: Porto Seguro, Parque Nacional Monte Pascoal, Sede, 150 m, 12–23/v/2021, 16°53′16.13″S 39°24′57.46″W, Simeão Moraes leg., Pasi Sihvonen, prep. number 2868; Pasi Sihvonen, DNA sample 1544, specimen ID http://id.luomus.fi/GBT.13 (ZMH).

# DISCUSSION

## New species

The two species described herein have idiosyncratic wing patterns, distinct from other lichen moth species in the Atlantic Forest biome (SS Moraes, pers. obs., 2021). Although the geographical distribution recorded is narrow, we expect that these species might be found in other areas in similar habitats near to Rio de Janeiro and Porto Seguro. These species may have been overlooked because of their small size, and secondly, the specialists investigating the Neotropical fauna of lichen moths are scarce.

The here described taxon *heliae* was assigned to the genus *Nodozana* Druce, based on DNA barcode data (see above) and similar wing pattern with *Nodozana toulgoeti*, but we highlight that genus-level systematics of Neotropical lichen moths are poorly resolved, and more research is needed. *N. toulgoeti* is known for French Guiana. For the taxon *pataxo*, the forewing pattern with white/beige background and black maculae/dots are shared with species in the genus *Epeiromulona* Field, in which it was assigned.

## Sclerotized and membranous structures

In Lepidoptera taxonomy, wing venation characters, in addition to abdomen and genitalia characters, are among the most diagnostic, routinely studied, and illustrated in scientific publications (*e.g.*, *Carter & Kristensen, 1998*; *Winter, 2000*; *Moraes & Duarte, 2009*). However, because Lepidoptera are covered with scales, these characters cannot be studied without scale removal. The widely used protocols are partly destructive, such as wing bleaching that removes color from scales (*e.g.*, *Moraes & Duarte, 2009*; *Murillo-Ramos, Hernández-Mejía & Llorente-Bousquets, 2016*) or KOH approach that dissolves fat but makes the sclerotized structures in the abdomen and genitalia visible (*e.g.*, *Hardwick, 1950*; *Robinson, 1976*). Although membranous structures may be visible to some degree after KOH treatment, their visualization also depends on the successful application of stains such as Chlorazol Black and Eosin. In these approaches, some membranous structures such as ducts, are routinely removed, and other abdominal structures, such as androconial scales and pheromone glands, are rarely illustrated.

We noted that non-destructive micro-CT imaging and post-processing of 3D models provided relatively easy and informative access to specific sclerotized and non-sclerotized structures. The 3D wing models made it possible to identify veins in a detailed manner and folds on the wing membrane close to the anal veins in both wings. Sometimes these folds are mistakenly identified as veins on 2D images. For the male genitalia, 3D models clearly illustrated the majority of the sclerotized structures, such as tegumen, uncus, transtilla, phallus, and valvae (Fig. 6). Non-sclerotized structures were difficult to visualize, including for instance membranous juxta in the male genitalia of *Nodozana heliae* Moraes **sp. nov.** and the membranous structures in the female genitalia (Fig. 7). For the female genitalia, which are primarily membranous, most of the structures could be identified only *via* the sequential study of the sagittal slices (Fig. 7) by using sclerotized structures such as signa, antrum, and ductus bursae as anchor points, and from those structures to infer the outer surface of the membranous corpus bursae and ductus bursae (Fig. 7). By carefully adjusting the 3D histogram it was possible to identify other regions with very low density, representing the interior of the corpus bursae, membranous ducts, and margins of the pheromone glands (Fig. 7).

## Muscles

Four genital muscles were clearly visible in the 3D models (Fig. 6, Supplementary Material 2): m1, m5 (7), m6 (5) and m7 (6). Because micro-CT models illustrate the structures in their natural position, it allowed for inferring the precise origin and inserting regions of muscle fibers and their naming. For example, the anatomical position of the phallus on the genital capsule allowed the distinction between the protractor and retractor muscles of the phallus (m6 (5) and m7 (6)) (Fig. 6). The former appears longer, with origin in the dorsomedial portion of the vinculum and the insertion in the base of the phallus, close to the caecum; the latter set of muscles are shorter and originates in the most ventral portion of the vinculum (saccus) and inserts in the central-ventral portion of the phallus. We did not detect the presence of muscle m8 (3), which originates in the median part of the vinculum and inserts at the distal margin of the juxta, being an indirect abductor of the

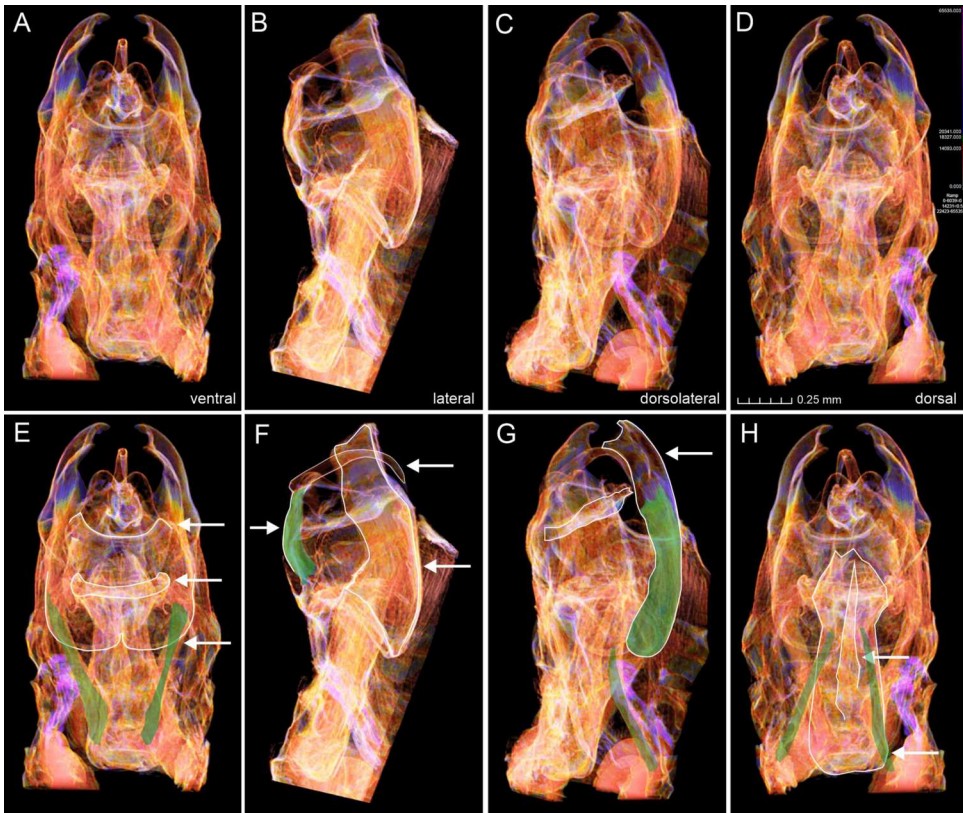

**Figure 6 Micro-CT images of male genitalia of *Nodozana heliae* Moraes sp. nov. (paratype).** (A–D)
Images from the 3D model in different angles. (E–F) Structures from different angles as indicated. (E)
Posterior margin of 8th sternite, transtilla, anterior margin of valva highlighted, protractors muscles of
aedeagus marked in green. (F) Margins of uncus, tegumen and valva highlighted, flexor muscle of uncus
marked with green. (G) Margin of valva and subscaphium highlighted, flexor muscle of valva and retrac-
tor muscle of aedeagus marked with green. (H) Margins of aedeagus and cornutus highlighted, retractors
muscles of aedeagus marked in green.

valvae (*Kuznetzov et al., 2004*). The absence of m8 (3) might be correlated with the lack of
a sclerotized juxta in the *Nodozana heliae* Moraes sp. nov.

The abductor and adductor musculature of the valva m3 (2) and m4 were not clearly
distinguished in the 3D models. Although it is possible to identify something similar
to muscles in the region between the transtilla and the valva, a better rendering of the
3D models is necessary to accurately assess the presence of these muscles, as well as the
retractor muscle of the vesica (m21), usually located inside the base of phallus (*Kuznetzov
et al., 2004*). In our study the valvae of the new species are morphologically simple, without
all subdivisions proposed by *Sibatani et al. (1954)*, but we identified the intravalval muscle
m5 (7), the flexor of clasper, which indicates that it may be possible also to identify more
muscles in species with more complex valva, such as some lichen moths species in the
genus *Inopsis* Felder, 1874 in which it is possible to identify all six subdivisions of valva.
Further, the micro-CT approach may help to access the configuration of the intravalval
musculature and to investigate whether the valva subunits have intrinsic musculature.

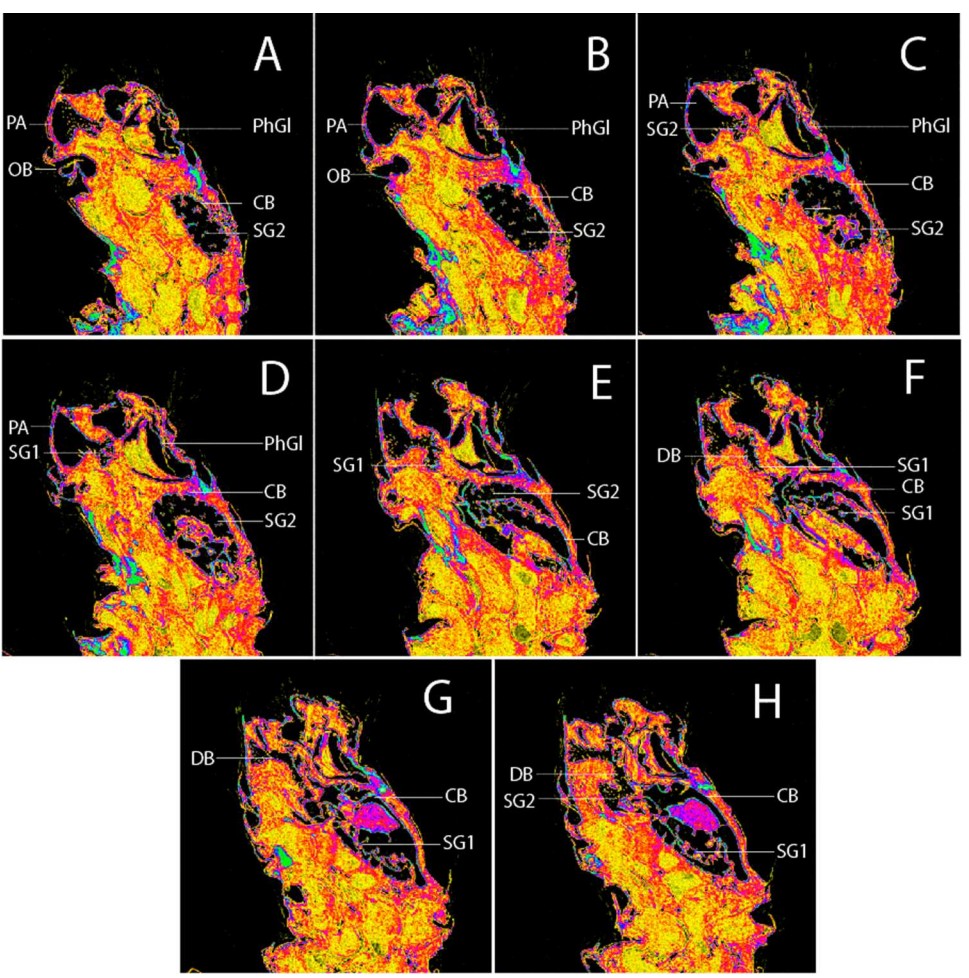

**Figure 7 Sagittal slices from 3D model of female genitalia of *Nodozana heliae* Moraes sp. nov. (paratype).** (A–H) Slices showing margin of sclerotized and membranous structures. CB, corpus bursae; DB, ductus bursae; PA, papillae anales; PhGl, pheromone gland; OB, ostium bursae; SG1, signum 1; SG2, signum 2.

Additionally, the micro-CT may provide new evidence on serial homology studies for the appendicular origin of the male genitalia and allow the evaluation of plesiomorphic or apomorphic state in a phylogenetic context (*Moraes, 2014*).

## Classical approaches are superior for specific structures

The classical, partly destructive dissection methods allowed us to identify some membranous structures in better detail (Figs. 2–5). These include, for instance, the abdominal skin and specialized scales on the male 8th sternite in *Nodozana heliae* Moraes **sp. nov.**, the ductus ejaculatorius in males, and the ductus bursae, corpus bursae and papillae anales in females. Details of the male vesica, *i.e.,* shape, size, and position of sclerotized structures and membranous diverticula, were only visible in the classical approach (*Sihvonen, 2001*) (Figs. 2 and 4). In lichen moths, the vesica is somewhat complex and very informative for taxonomy, with some species showing several diverticula

and different shapes of cornuti (*Durante & Apinda-Legnouo, 2022*; *Volynkin et al., 2022a*; *Volynkin, Cerny & De Vos, 2022b*). Our results support the earlier view: also, in the smallest lichen moths, the vesica is complex and contains diagnostic characters.

## CONCLUSIONS

Our results demonstrate that micro-CT scanning combined with traditional dissection protocols can create virtual dissections of the male genitalia in lichen moths, where most of the diagnostic structures are visible. Furthermore, 3D reconstructions have the advantage of visualizing the morphological structures, such as the genitalia muscles and wing venation, without scale removal. Muscle information is usually lost with KOH, and wing colors are lost if bleaching is used.

Although the 3D reconstructions presented here are promising, we emphasize that micro-CT scanning cannot fully replace the abdomen and genitalia dissections in Lepidoptera for systematics and taxonomy purposes. Many specimens will not produce satisfactory 3D models, and membranous structures embedded inside soft tissue such as body fat, appear problematic. One example is the female genitalia in Lepidoptera, which is almost entirely internal and membranous. Another example comes from some critical traits in the male genitalia, such as the vesica. The latter's shape and number of lobes can only be fully understood when the structure is physically everted and shown as maximally inflated. Wings are somewhat 2D structures, and for the study of wing venation, we recommend micro-CT scanning as the first approach. We also acknowledge that further post-image processing of raw data could allow identifying additional structures, which were not visible to us.

The advantages of using micro-CT in systematics are undeniable. First, it represents a non-destructive method that can study the type specimens and/or rare species. Second, it has the potential to access information on the morphology and the functional anatomy of structures in their natural anatomical position, which otherwise would be deformed or lost with classical dissection protocols. The use of micro-CT offers new opportunities for enhancing taxonomic descriptions and comparative studies, *e.g.*, *via* video files, broadening the utility of morphological characters also in other disciplines in biology.

### Abbreviations

| | |
|---|---|
| **MZUSP** | Museu de Zoologia da Universidade de São Paulo, Brazil |
| **ZMH** | Finnish Museum of Natural History, University of Helsinki, Finland |
| **ZUEC** | Zoological collection, Museu de Diversidade Biológica, Universidade Estadual de Campinas, Brazil |

## ACKNOWLEDGEMENTS

Elina Laiho (Finnish Museum of Natural History, University of Helsinki) and Eduardo de Proença Barbosa are thanked for processing the DNA barcodes. The present study is registered in the SISGEN (A6751E2).

### Funding

Simeão de Souza Moraes was supported by the European Commission (grant Synthesys+ FI-TAF-TA3-002) and FAPESP (2015/17047–5, 2016/20196–5). André Victor Lucci Freitas was supported by FAPESP (2021/03868-8), and the Brazilian Research Council–CNPq (421248/2017-3 and 304291/2020-0). Pasi Sihvonen received a grant from the Research Council of Finland (decision 331995). The funders had no role in study design, data collection and analysis, decision to publish, or preparation of the manuscript.

### Grant Disclosures

The following grant information was disclosed by the authors:
European Commission: Synthesys+ FI-TAF-TA3-002.
FAPESP: 2015/17047–5, 2016/20196–5, 2021/03868-8.
Brazilian Research Council–CNPq: 421248/2017-3, 304291/2020-0.
Research Council of Finland: 331995.

### Competing Interests

The authors declare there are no competing interests.

### Author Contributions

- Simeão S. Moraes conceived and designed the experiments, performed the experiments, analyzed the data, prepared figures and/or tables, authored or reviewed drafts of the article, and approved the final draft.
- Max S. Söderholm performed the experiments, analyzed the data, authored or reviewed drafts of the article, generated the images from the micro-CT scan, and approved the final draft.
- Tamara M.C. Aguiar performed the experiments, authored or reviewed drafts of the article, and approved the final draft.
- André V.L. Freitas performed the experiments, authored or reviewed drafts of the article, and approved the final draft.
- Pasi Sihvonen conceived and designed the experiments, performed the experiments, analyzed the data, prepared figures and/or tables, authored or reviewed drafts of the article, photographed the specimens, and approved the final draft.

### DNA Deposition

The following information was supplied regarding the deposition of DNA sequences:
The sequences are available at GenBank: OQ029493 and OQ947288.

### Data Availability

The 3D videos obtained with Micro-CT and CT scans are available at Morphosource:
- Media 000488436: Male Genitalia: https://doi.org/10.17602/M2/M488436
- Media 000488429: Wing Venation: https://doi.org/10.17602/M2/M488429

- Media 000488448: Female Genitalia: https://doi.org/10.17602/M2/M488448
- Media 000488442: Female Genitalia: https://doi.org/10.17602/M2/M488442
- Media 000529976: Male Genitalia: https://doi.org/10.17602/M2/M529976
- Media 000529968: Female Genitalia: https://doi.org/10.17602/M2/M529968
- Media 000529972: Wing Venation: https://doi.org/10.17602/M2/M529972

## New Species Registration

The following information was supplied regarding the registration of a newly described species:

Publication LSID: urn:lsid:zoobank.org:pub:68906FAC-208D-48D7-B69C-4ABDE6CF A0D6

*Nodozana heliae*: urn:lsid:zoobank.org:act:F97C4D7C-65A3-4EEC-8D34-F84D5C7346EE

*Epeiromulona pataxo*: urn:lsid:zoobank.org:act:D4637E11-169E-45D0-8A25-6A61F68939A5

## Supplemental Information

Supplemental information for this article can be found online at http://dx.doi.org/10.7717/peerj.15505#supplemental-information.

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
