# Peer review of "Micro-CT imaging in species description: exploring beyond sclerotized structures in lichen moths (Lepidoptera: Erebidae, Arctiinae, Lithosiini)"

_PeerJ, doi:10.7717/peerj.15505_

## Round 0.1 · original submission · Minor Revisions

Dear Authors, referees positively consider your submission. My suggestion is to consider the notes that are given to ameliorate the draft. I am ready to consider the draft after the resubmission.

As your editor, I was pleased to review your text and upload the .pdf with my suggestions. I hope to evaluate the new version of your work very soon.
Francesco

·

Basic reporting

I’m not a native speaker in English but the language of the manuscript seems adequate. Only small typos were detected, and some sentences should be better writing to make clear the main idea.
Literature used is appropriated, but I noticed that some parts of the text are missing references to support the rationality. I also made recommendation of some works with the intention to help improving the discussion.
Although the journal does not restrict the choice of reference style, it is mandatory that authors list them in a consistent and standardized way in the manuscript. I detected several inconsistences, for instance, in some it was used a comma to separate the number of the journal volume and pages, in others the journal is not italicized, etc. I recommend the authors carefully review and fix them. DOI references are also missing. Also, some references listed are not cited in the text.
Figures are well-informative and presented in high-quality. The only problem I detected was the missing scale bars in figures 2, 3, 4 and 5. I strongly recommend the authors to add them. Legends of the figures are fine and adequately descriptive. Videos in the supplementary material are also informative and helps to better visualize the descriptions provided in the text.

Experimental design

In my opinion, the work meets the aims and scope of the journal. It is a high relevant and well-delineated study contrasting the use of traditional dissection methods with micro-CT virtual dissection tool. To the best of my knowledge, it is the first study of this type in Lepidoptera, mainly by exploring features in muscles and female genitalia. Additionally, it is a very good example of the use of distinct approaches for species delimitation. Other merit of the study is using specimens from Brazil, one of the major biodiversity hotspot in the world, which still suffers from the lack of taxonomists for Lepidoptera, basic information for several groups is scarce and there are a lot of species to be discovered and described.
The methods used are adequate for the purpose of the study and most of them are well-descripted allowing the reproducibility in future studies. I only recommend adding information on the protocols used for traditional dissection of the male and female genitalia.

Validity of the findings

The description of the new species meets ICZN standards as well as the ‘new species policy’ of the journal. Evidence presented in the study strongly supports the finds. However, etymology of both species should be improved. In a general context, my only comment is that, considering both new species were described also based on females, why do the authors not explored the female wing venation in micro-CT or even using traditional diaphanization technique? As several lepidopteran groups are sexually dimorphic regarding the wing venation, especially the humeral and disco cellular veins, etc., adding this information and respective illustrations could be informative for the systematics of the group. Conclusions are well stated but could be also improved.

Additional comments

All the comments and suggestions mentioned above (and other minors one) are detailed and highlighted in the attached PDF.

Reviewer 2 ·

Basic reporting

Even if this research does not represent an improvement in the field of micro-CT application for the study of morphology and anatomy of insects, I did find it very interesting since It would aim to highlight the advantages that such technique offers as a non-destructive technique when studying very rare specimen. I believe therefore that this aspect should be more emphasized over the paper.

Below are some suggestions especially concerning microCT analysis and related results.

From line 165
It is not really clear to me whether you used any kind of sample preparation before CT analysis (e.g. fixation, drying…) or not. In the latter case, it would be useful to highlight and clarify the reasons of your choice (maybe to simulate the same condition of working with a very rare specimen as you mentioned in the introduction?). However, drying is crucial in micro-CT and allows less noisy and more defined acquisition.

Line 174
Which was the angular acquisition step?

Line 183
In this regard, you should use a color-bar for all of your CT pseudo-colored renderings in order to let readers understand density variations (actually, also other images deserve a metric scale).

Line 215
Wings images in Fig.1 are surely very beautiful and effective. However, it would be important to highlight why and how using micro-CT can be better than traditional approaches: for instance, beside those (2D) pictures, it would be nice to show also some 3D renderings or even better a 3D-model of the only venation (you could do this via segmentation).

Line 221
About Figure 6, probably the details highlighted in figure E, F, G and H can be shown directly in A, B, C and D. Rather, It would be interesting to show also images from other points of view, e.g. a frontal view of the back side of the genitalia structure.

Line 238
In this regard, some sample preparation would have been helpful

Line 244
About Figure 7, beside the need of a color-bar as mentioned before, these images are really noisy and confused. You could try to improve the quality of these images; for instance, it seems that yellow regions (low density) are not useful to show the structures you’re interested to. Maybe deselecting those regions from the attenuation histogram you could enhance the contrast of other ones.
Furthermore, Figure 7 needs to be better described in the caption, e.g. clarifying that A is a peripheral section and from A to H you move inward your sample.

Experimental design

No comment

Validity of the findings

No comment

Reviewer 3 ·

Basic reporting

In this manuscript the authors describe two new species of lichen moths using a very modern approach with micro-CT scanning of specimens. The authors compare the results from this scanning with the traditional approach to the examination of the morphology and they make a good case that this method is in a number of ways an improvement.

Overall the work described is detailed and careful and I have no comments on the content relating to the description. The language is clear and easy to understand; however, it needs to be cleaned up by a fluent English speaker because there are issues especially with respect to the usage of prepositions and articles.

It would be nice if the authors could include a short paragraph outlining how long it takes to make these micro-CT scans and process the data, and compare this with the time that goes into the traditional preparation of wings for wing vein examination as well as genitalic dissections. I understand that saving the time is not necessarily the main issue since this method can be applied non-destructively, but it would be good to have some time frame for comparison.

All of the figures are beautiful!

I made a number of small corrections on the pdf.

Experimental design

No comment.

Validity of the findings

No comment.

Additional comments

No comment.

Annotated reviews are not available for download in order to protect the identity of reviewers who chose to remain anonymous.

---

## Round 0.2 · accepted · Accept

Dear authors,

Thank you for addressing the most significant part of the reviewers' comments. A pdf. follows this letter with very few suggestions from your comments to one of the referees. The paper participates in a research line using µCT to unveil details in arthropod morphology, either of functional or taxonomic significance.
I hope this will support an excellent research level in entomology.